# SOP-Agent: Empower General Purpose AI Agent with Domain-Specific SOPs

## Abstract

Despite significant advancements in general-purpose AI agents, several challenges still hinder their practical application in real-world scenarios. First, the limited planning capabilities of Large Language Models (LLM) restrict AI agents from effectively solving complex tasks that require long-horizon planning (Liu et al., 2023). Second, general-purpose AI agents struggle to efficiently utilize domain-specific knowledge and human expertise. In this paper, we introduce the Standard Operational Procedure-guided Agent (SOP-agent), a novel framework for constructing domain-specific agents through pseudocode-style Standard Operational Procedures (SOPs) written in natural language. Formally, we represent a SOP as a decision graph, which is traversed to guide the agent in completing tasks specified by the SOP. We conduct extensive experiments across tasks in multiple domains, including decision-making, search and reasoning, code generation, data cleaning, and grounded customer service. The SOP-agent demonstrates excellent versatility, achieving performance superior to general-purpose agent frameworks and comparable to domain-specific agent systems. Additionally, we introduce the Grounded Customer Service Benchmark, the first benchmark designed to evaluate the grounded decision-making capabilities of AI agents in customer service scenarios based on SOPs.

## 1 Introduction

Autonomous general purpose agents, built on the capabilities of Large Language Models (LLMs), have shown remarkable potential in performing a wide range of tasks. Existing general purpose agent systems (Wu et al. (2024), Yang et al. (2023a), Yao et al. (2023b), Li et al. (2023), Zhang et al. (2024c), Chen et al. (2024), Mei et al. (2024), Chase & contributors (2024), Nakajima (2023), Team (2023)) made significant process in fields such as planning (Yao et al. (2023b), Shinn et al. (2023), Wei et al. (2023), Yao et al. (2023a)), memory optimization (Mei et al. (2024), Zhao et al. (2023), Zhong et al. (2023), Liu et al. (2024)), tool calling (Zheng et al. (2024)), multi-agent cooperation (Wu et al. (2024), Yang et al. (2023a), Li et al. (2023)). However, their applications in the real world remain limited due to several fundamental challenges. Chief among these are the shortcomings in planning capabilities, LLMs generated plans suffer from hallucinations, and lack of feasibility and efficiency, adding to that agents usually do not have efficient tools to perform fine-grained evaluation of the plan (Huang et al., 2024c). Besides, LLMs are not reliable in solving long-horizon planning tasks (Liu et al., 2023), making planning a significant challenge in real-world applications. Moreover, few works have explored how to integrate domain-specific knowledge and human experience with AI agents, which is essential for more specialized real-world applications.

While general-purpose agents demonstrate great versatility, the complexity of real-world tasks necessitates the development of more specialized, domain-specific agents. These agents are designed to excel in targeted areas by incorporating deeper domain expertise, complex workflows, and task-specific optimizations. For example, domain-specific AI systems MetaGPT (Hong et al., 2023), which is tailored for programming tasks, not only harness the general reasoning abilities of large language models (LLMs) but also integrate widely adopted software development SOPs into its multi-agent framework to enhance precision in programming. Similarly, AutoCrawler (Huang et al., 2024b) incorporates a domain-specific workflow to leverage the hierarchical structure of HTML data to progressively understand web content.

Generally, domain-specific agents (Hong et al. (2023), Huang et al. (2024a), Huang et al. (2024b), Gao et al. (2024), Ghafarollahi & Buehler (2024)) rely on workflows designed based on human experience, often hardcoded, to improve their performance on fixed tasks. However, hardcoding human-designed workflows to build domain-specific agents is only economical for high-demand tasks, such as programming. In practice, different applications require different SOPs, even within the same domain, SOPs may vary a lot from company to company. Moreover, SOPs are constantly evolving, which further makes building traditional domain-specific agents impractical and unscalable.

To tackle these challenges, we propose a novel framework: the Standard Operational Procedure-guided Agent (SOP-agent). Our approach integrates the flexibility of general-purpose AI agents with the benefits of a domain-specific workflow designed based on human intelligence and experience. By utilizing pseudocode-style SOPs written in natural language, the SOP-agent navigates task execution by selectively traversing a decision graph, offering structured, comprehensible instructions to guide the agent's behaviors. We also limit the agent's accessible tools to a filtered set based on the SOP.

We conduct an empirical evaluation of our SOP-agent, comparing it with strong baselines, including state-of-the-art domain-specific agents. Our evaluation covers a diverse range of topics, demonstrating the high versatility of the SOP-agent. Guided by a well-designed SOP, the SOP-agent outperforms AutoGPT by 66.2% in a zero-shot setting on the ALFWorld benchmark (Shridhar et al., 2020). The SOP-agent achieves competitive Pass@1 scores on both the HumanEval (Chen et al., 2021) benchmark (86.6) and the MBPP (Austin et al., 2021) benchmark (89.5), compared to domain-specific methods. Additionally, we test the agents' ability in data cleaning, a complex real-world task that requires domain knowledge. Our system achieves a 100% success rate, significantly higher than AutoGPT (87.5%) and comparable to the state-of-the-art domain-specific agent MetaGPT's Data Interpreter (Hong et al., 2024) in solving data-driven tasks. Inspired by prompt engineering, we propose improving the SOP-agent's robustness through a process we term SOP engineering. We also introduce a benchmark dataset specifically designed to evaluate the agent's grounded decision-making abilities in customer service contexts, where our system achieves an impressive accuracy of 99.8%.

The key contributions of this paper are threefold.

- First, we present the SOP-agent framework, the first system, to the best of our knowledge, for building complex domain-specific agents with natural language workflow.

- Second, we introduce an evaluation benchmark tailored to measure the efficacy of AI agents in performing extensive grounded decision-making in customer service scenarios.

- Third, our experiments on *Grounded Customer Service Benchmark* show that our SOP agent can achieve both high robustness and accuracy through SOP engineering.

## 2 BACKGROUND AND RELATED WORK

**Use of Human SOP in Domain-specific Agents** Many domain-specific AI agent systems use human-designed SOP to optimize specific tasks. In code generation, most existing programming agents (Huang et al. (2024a), Qian et al. (2024), Hong et al. (2023), Wang et al. (2024), Zhang et al. (2024b), Yang et al. (2024)) use predefined debugging loop for self-debugging (Chen et al., 2023). Besides, MetaGPT introduced by Hong et al. (2023) hardcodes a software development SOP that involves cascaded action execution of different agents (e.g., product manager, engineer...), the SOP also controls communication between agents. In the CodeAgent (Zhang et al., 2024b), a set of rules is applied to establish the proper sequence for tool usage, ensuring that thorough research, including web searches and document reading, is conducted before coding. In other domains, Gao et al. (2024) proposed an AI system capable of automatically conducting biological research. This system utilizes a "self-driving lab", where actions, including hypothesis generation, experiment design, conducting experiments, and analyzing experiment results, are performed in cycles. In another task of building a web crawler for web content understanding, Huang et al. (2024b) developed AutoCrawler, which implements a human SOP to recursively search for relevant information through a two-stage process that traverses the webpage's DOM tree.

**Rule-based Expert System** Rule-based expert system (ES), one of the earliest attempts made in AI, was first introduced by Lindsay et al. (1993) to solve a scientific hypothesis formation problem with a knowledge-driven approach. Later, Shortliffe (1977) proposed the IF-THEN heuristic rule, which later became a paradigm in Rule-based ES design. Our SOP-agent resembles the Mycin system as we adopt its IF-THEN formula and power it with LLM's reasoning ability.

**Grounded Agents and Language Models** Few existing works ground AI agents on predefined workflows. We found AutoGPT+P (Birr et al. (2024)), which combines an affordance-based scene representation with a symbolic planner designed specifically for embodied robotic tasks. Similar to our work, Roy et al. (2024) introduces a Flow-Adhering Planning algorithm (FLAP), in which a set of predefined plans are provided to the agent in textual format to provide domain-specific knowledge. Each plan is a sequence of actions (flow) that needs to be executed sequentially. Additionally, Qiao et al. (2024) proposes to use a world-knowledge model, trained on the action trajectories collected from the simulation environment to affect the agent's behavior. In the research direction of grounded LLM, Xie et al. (2023) uses LLM to translate plans written in natural language to executable Planning Domain Definition Language (PDDL). Later researchers (Liu et al. (2023), Yang et al. (2023b), Dagan et al. (2023)) further use symbolic planners or simulators to execute LLM-translated symbolic plans or simulation scripts and ground LLM with the simulation results. Zhang et al. (2024a) use learned heuristics to guide the logical graph search to select the next action from a set of admissible actions. Although existing works (Roy et al. (2024), Birr et al. (2024), Liu et al. (2023), Yang et al. (2023b), Dagan et al. (2023)), has explored how to ground agent/LLM's output on predefined workflows, there still lacks a method that can handle complex workflow management, such as branching and looping, without simulation environments or planners.

## 3 METHOD

We propose to track the state of the agent in workflows and dynamically adapt a plan based on observation through selective depth-first-search (DFS) traversal of the decision graph. The overall design, as depicted in Figure 1, can provide SOP guidance to existing agents, such as Act and ReAct, to make the agent follow the workflow. For clarity, in the rest of this paper, we define two key concepts: (1) **Action**: A semantic representation of a task or behavior, such as "read a book." (2) **Function call**: An executable program that acts, often parameterized, such as read(obj="book").

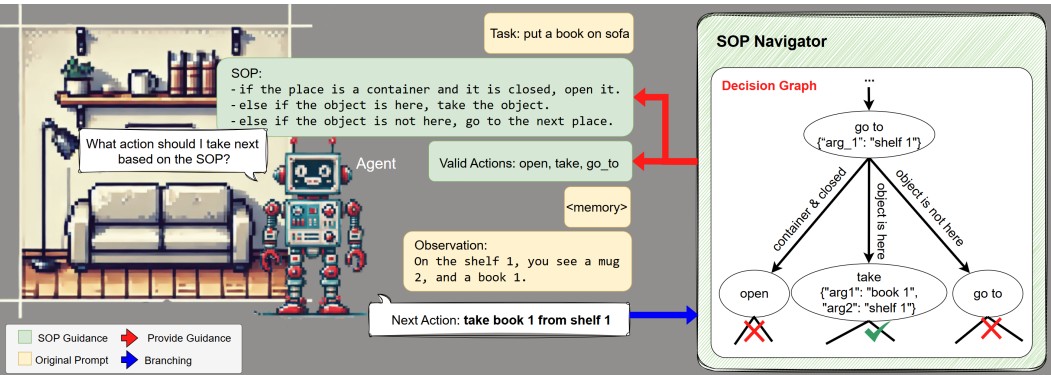

Figure 1: Left: **The SOP-Agent framework.** During each step, the SOP-Navigator formats a textual SOP and provides a filtered set of valid actions to guide the base agent's behavior. The agent needs to generate the next action, which is used to traverse the decision graph and update the state of the SOP-Navigator. Right: **The Decision Graph.** The figure shows a segment from a decision graph.

### 3.1 STANDARD OPERATING PROCEDURE (SOP)

We represent the SOPs as decision graphs, where each node signifies a candidate action at the current step. These actions can influence the environment, allowing the system to actively gather additional

evidence for future decision-making on demand. Each edge corresponds to a *IF* condition or an unconditional *ALWAYS* condition. The textitALWAYS condition implies that the subsequent action will always be executed. A node can have multiple directed edges connecting it to its sub-nodes, and the condition on each edge need not be mutually exclusive, meaning that any sub-tree that meets the condition will be traversed. This simple yet efficient design enables core features such as cascaded execution of tasks, conditional branching, and looping, providing users with the flexibility to design complex workflows using pseudocode-style SOPs written in natural language.

### 3.2 SOP Guidance

Our approach to guiding agent's behavior through SOPs consists of two components. First, the conditions and actions of subnodes are formatted into structural prompts to guide the agent's behavior. Second, we provide the agent with a filtered set of valid function callings (see Figure 1). These function callings are restricted to those associated with subnodes, which effectively limits the action space and improves decision-making robustness.

### 3.3 Branching And Traversing On The Decision Graph

**Branching**    To selectively traverse the decision graph, a common approach is to first determine whether the condition for each node is met. If the condition is met, the corresponding function (if any) will be executed with parameters generated by separate LLM calls, resulting in $1 + |branches\_with\_function\_calls|$ queries. However, this approach can be optimized in certain scenarios where the function callings of each node are different. In this case, the function callings that the agent made help determine which conditions are met, allowing for more efficient branching. We use OpenAI's GPT-4, which provides a tool call interface that supports generating all the necessary function calls in a single query. We explore each subbranch based on the selected function call in DFS fashion as shown in Figure 1. This approach reduces the number of required queries to one per traversal. For more details on the cost analysis, refer to Appendix A. There are two scenarios in which actions cannot be distinguished: (1) when a node has at least two sub-nodes that perform the same function calling, and (2) when a node has at least two sub-nodes that do not perform any function calling. In these cases, we have to use the naive approach as described at the beginning of this paragraph. We prompt the LLM to call all applicable functions from predefined "dummy" function calls like "explore_subtree_A" and "explore_subtree_B". Afterward, the LLM generates the actual actions during the second phase of traversal. See Appendix B for more details.

**DFS-based Selective Traversing**    We employ DFS to traverse the decision graph selectively. On each step, we use the branching mechanism as stated above to select branches whose preconditions are met based on observation. Then, we recursively perform DFS on selected sub-branches.

## 4 EXPERIMENTS

We evaluate SOP-Agent across four domains to assess its versatility: (1) decision making, (2) multi-hop question answering via interactive searching, and (3) code generation, and (4) data cleaning.

### 4.1 Decision Making

**Experimental setup**    *ALFWorld* (Shridhar et al., 2020) is a virtual, text-based game built on the ALFRED benchmark dataset (Shridhar et al., 2019). ALFWorld provides a simulator that simulates six types of household-related tasks, including (1) put sth. in/on sth./spl., (2) find sth., heat it then put it in/on sth./spl., (3) find sth., cool it then put it in/on sth./spl., (4) find sth., clean it then put it in/on sth./spl., (5) examine sth. under a desklamp, (6) take and put sth. in/on sth./spl. twice. In our experiments, we use the existing ALFWorld simulator, which provides eight admissible actions: go to, take, put, heat, cool, open, clean, and use. Since the ALFWorld environment contains more than 50 possible locations, efficient exploration requires a targeted search strategy. For example, to search for an object, start with the location where the object is most likely to appear, then iteratively explore other locations. We manually write an SOP using human-designed optimal strategies for all six tasks. The SOP can be found in Appendix F. For the base agent, we choose to use a ReAct agent because the action trajectory contains useful information in the ALFWorld task.

**Baselines** For comparison, we evaluate the performance of the SOP-guided agent against Auto-GPT (Yang et al., 2023a) and the original ReAct Agent. The experimental results for AutoGPT are based on the data reported in the original AutoGPT paper. To ensure a fair comparison, all agents were evaluated using GPT-4 on the same set of 134 unseen tests with a low-temperature setting to minimize randomness in responses (SOP-Agent: 0.0, AutoGPT: 0.01, ReAct: 0.0). For the two experiments that use few-shot prompting, we use identical few-shot prompts generated by the official evaluation script of ReAct. For the SOP-agent and ReAct experiments, we limit the number of GPT calls to 50. Furthermore, AutoGPT also reports the performance of a variant that incorporates an imitation learning (IL) model, trained using expert demonstrations.

**Evaluation Metrics** For evaluation metrics, we use the success rate, $success\_rate = \frac{number\_of\_success\_trial}{number\_total\_trial}$. The trial is successful if the ALFWorld game simulator returns a success signal before the agent terminates, crashes, or reaches the maximum GPT call limit.

Table 1: Agents' Performance on ALFWorld

| Model | Success Rate |
|---|---|
| ReAct(GPT4, few-shot) | 0.843 |
| Auto-GPT(GPT4, zero-shot) | 0.485 |
| Auto-GPT(GPT4, zero-shot) + IL | 0.515 |
| SOP-Agent(GPT4, zero-shot) | 0.806 |
| **SOP-Agent(GPT4, few-shot)** | **0.888** |

Table 2: Detailed Success Rates By Task Categories

| Task | ReAct (few-shot) | SOP-Agent (few-shot) | SOP-Agent (zero-shot) |
|---|---|---|---|
| put sth. in/on sth./spl. | 0.880 | **1.000** | 0.958 |
| find sth., clean it then put it in/on sth./spl. | **0.938** | 0.903 | 0.935 |
| find sth., heat it then put it in/on sth./spl. | 0.727 | 0.826 | **0.913** |
| find sth., cool it then put it in/on sth./spl. | 0.952 | **0.952** | 0.810 |
| examine sth. under a desklamp | 0.765 | **0.778** | 0.556 |
| take sth. and put them in/on sth./spl. twice | 0.706 | **0.824** | 0.470 |

**RESULTS AND OBSERVATIONS** Table 1 shows the performance of different agent frameworks on the ALFWorld benchmark. Detailed breakdown of success rate by task categories of the ReAct and SOP-agent experiments are listed in Table 2. Under a few-shot setting, our system performs better in five out of six takes and achieves an overall success rate that is 4.5% higher than ReAct. Compared with AutoGPT under zero-shot setting, with the help of human-crafted SOPs, our system significantly outperforms AutoGPT, even beats the variant with imitation learning model by a large margin (66.2% improvement on AutoGPT and 56.5% improvement on its IL variant).

While the SOP-agent achieves a remarkable overall success rate and very high success rate (greater than 90 percent) on certain task categories, we also observe that it doesn't perform robustly on the last two tasks. Through manual examination of the action trajectory, we found that sometimes the LLM doesn't follow the SOP and performs actions based on its internal knowledge, causing the system to fail. For example, in the "examine sth. under a desklamp" task, the agent is prone to take the desklamp despite that the SOP specifically instructs it to take the object to be examined.

## 4.2 MULTI-HOP QUESTION ANSWERING VIA INTERACTIVE SEARCHING

**EXPERIMENTAL SETUP** We utilize HotpotQA (Yang et al., 2018) to evaluate the agents' ability to perform interactive searching and multi-hop reasoning. HotpotQA is a task designed for multihop question answering, where an agent iteratively searches Wikipedia passages to gather

information and answer questions. Each question requires information from at least two distinct Wikipedia passages. The agent interacts with a search engine through three actions: (1) search[entity]: This action searches for an exactly matched entity and retrieves the corresponding passage from the Wikipedia database. If no exact match is found, it returns a list of similar entities. (2) lookup[keyword]: This action returns the next sentence that contains the specified keyword from the current passage. (3) finish[answer]: This action is used to submit the final answer to the question. Similar to the ALFWorld experiment, we adapt the ReAct agent by incorporating a Standard Operating Procedure (SOP), which provides step-by-step instructions on how to navigate the multi-hop searching and reasoning process. The manually crafted SOP for this task is detailed in Appendix F.

**Baselines** We compare the performance of the SOP-agent with that of the original ReAct agent. Both agents are evaluated under the same few-shot setting, with identical prompts. The experiments are conducted on the same set of 200 questions using GPT-4 with a temperature setting of 0.0.

**Evaluation Metrics** Following the ReAct paper, we use two metrics: (1) EM: the ratio of questions where the agent's response exactly matches the ground truth answer. (2) F-1 score: the F-1 score, which measures the average similarity between the agent's response with the ground-truth answer. We also analyze the difference in agents' behavior through an ablation study on several action patterns that we think can reflect agents' exploration abilities: (1) $total\_searches$: total number of search attempts, (2) $total\_lookups$: total number of lookup attempts, (3) $consecutive\_search\_same\_keywords$: the total number of search attempts using the same entity as the previous consecutive search attempt. (4) $consecutive\_search\_same\_keywords$: the total number of lookup attempts using the same keyword as the previous consecutive lookup attempt. (5)-(14) $lookup\_same\_keyword\_level\_N$: The total number of consecutive lookups using the same keyword at depth $N$, where $N$ represents the length of the consecutive lookup sequence. For example, the second lookup in $lookup[Taylor\ Swift] >> lookup[Taylor\ Swift]$ counts as a lookup at depth 2.

Table 3: Comparison of ReAct and SOP-Agent on Various Metrics

| Metrics | ReAct | SOP-Agent |
|---------|-------|-----------|
| EM | 0.448 | **0.464** |
| F-1 score | 0.589 | **0.609** |

Table 4: Ablation Study on Agents' Behavior Difference

| Metrics | ReAct | SOP-Agent | % of Change |
|---------|-------|-----------|-------------|
| total_searches | 572 | 590 | +3.15% |
| total_lookups | 104 | 107 | +2.88% |
| consecutive_search_same_keyword | 10 | 4 | -60.00% |
| consecutive_lookup_same_keyword | 28 | 50 | +78.57% |
| lookup_same_keyword_level_1 | 80 | 57 | -28.75% |
| lookup_same_keyword_level_2 | 11 | 14 | +27.27% |
| lookup_same_keyword_level_3 | 7 | 11 | +57.14% |
| lookup_same_keyword_level_4 | 4 | 9 | +125.00% |
| lookup_same_keyword_level_5 | 2 | 6 | +200.00% |
| lookup_same_keyword_level_6 | NA | 4 | $+\infty$ |
| lookup_same_keyword_level_7 | NA | 2 | $+\infty$ |
| lookup_same_keyword_level_8 | NA | 2 | $+\infty$ |
| lookup_same_keyword_level_9 | NA | 1 | $+\infty$ |
| lookup_same_keyword_level_10 | NA | 1 | $+\infty$ |

**RESULTS AND OBSERVATIONS** Despite that our experiment on HotpotQA only shows marginal improvements on ReAct in both metrics (+1.6% in EM and 0.02 up in F-1 score), (see Table 3), our ablation study results in Table 4, where positive changes are indicated in green text and

negative changes in red text, suggests that guiding agent with SOP noticeably shifts the action pattern of the base agent. First, SOP agents are less prone to search for the same entity multiple times, which is beneficial as searching for the same entity does not yield new observations. Second, the SOP-agent performs better in lookups, reflected by the increase in the depth of lookups, as ending the lookup before reaching the end of the article may risk missing important information.

## 4.3 CODE GENERATION

**Experimental Setup**   We use two widely adopted code generation benchmarks, HumanEval (Chen et al., 2021) and MBPP (Austin et al., 2021) to evaluate the code generation ability of the SOP-agent. To adapt to the code generation task, in both benchmarks, we guide a single Act agent with SOP that empowers the Act agent with debugging and self-reflection (Shinn et al., 2023) ability. Additionally, we incorporate a persistent, read-replace-only long-term memory. This allows the agent to see previously generated code, observations, and thoughts in the prompt for debugging and self-reflexion. For the HumanEval benchmark, we use the existing HumanEval evaluation harness that provides a testing environment for 164 coding tasks. For the MBPP dataset, we adopt the same evaluation setting as AgentCoder (Huang et al., 2024a) and use the test split of the sanitized subset of the MBPP dataset (257 data points) based on whether all provided unit test cases can pass. In both experiments, we use a temperature of 0.0. The SOPs used in both experiments can be found in Appendix G.

**Baselines**   For the HumanEval benchmark, we include baselines across different methodologies: large language models: (1) GPT-4 (0-shot), OctorCoder (GPT-4 with fine-tuned on coding tasks), coding systems: (1) Parsel (Zelikman et al., 2023), ANPL (Huang et al., 2023), agent systems: MetaGPT (Hong et al., 2023), L2MAC (Holt et al., 2024), MapCoder (Islam et al., 2024), Agent-Coder (Huang et al., 2024a). Among those baselines, MetaGPT, L2MAC, MapCoder, and Agent-Coder are multi-agent frameworks designed specifically for code generation tasks. For the MBPP benchmark, we compare our method with large language models: (1) GPT-4 (0-shot), (2) GPT-4 (few-shot), and agent system: (1) MapCoder (Islam et al., 2024), MetaGPT (Hong et al., 2023), AgentCoder (Huang et al., 2024a). For a fair comparison, all baselines use GPT-4 as the base LLM.

**Evaluation Metrics**   For both HumanEval and MBPP benchmark, we compare the Pass@1 score: ($Pass@1 = \frac{number\_passed\_tasks}{total\_task\_number}$) of different methods.

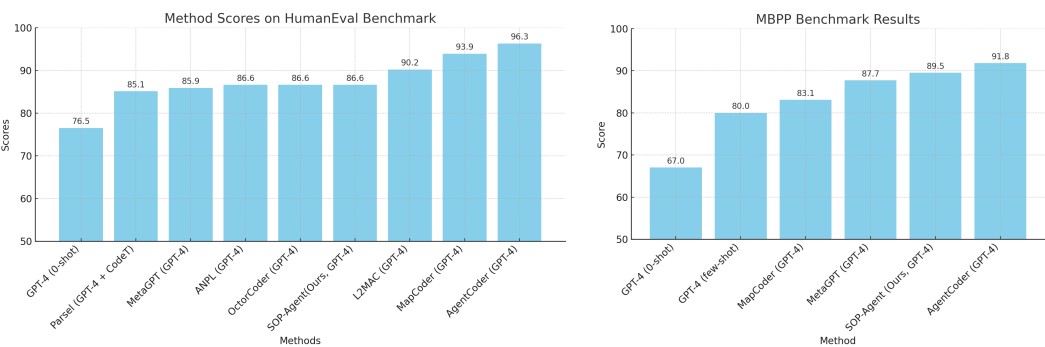

Figure 2: HumanEval benchmark results       Figure 3: MBPP benchmark results

**RESULTS AND OBSERVATIONS**   The evaluation results on the HumanEval benchmark (see Figure 2) and the MBPP benchmark (see Figure 3) demonstrate that the SOP-Agent, grounded with a code generation SOP, performs competitively on both the HumanEval and MBPP benchmarks compared to several strong domain-specific agent systems in coding. On the HumanEval benchmark, SOP-agent achieves a score of 86.6, which is better than MetaGPT and on par with OctoCoder and ANPL. On the MBPP benchmark, the SOP-Agent (GPT-4) achieves a score of 89.5, surpassing MapCoder (83.1) and MetaGPT (87.7). As the SOP agent gives a filtered toolset, clean and targeted instructions at each inference step, such as, "generate the code to..." and "think and reflect on...", it

is reasonable to treat the SOP-agent as a multi-agent system despite it is not grounded on agents' personas, which explains its superior performance in code generation.

## 4.4 DATA CLEANING

**Experimental Setup** To demonstrate that our proposed SOP agent can handle complex real-world problems in fields that requires specialized expertise with the help of external knowledge injected via SOPs, we test our agent framework in the scenario of data cleaning on 4 Kaggle challenge datasets. There include: (1) CO2 Emission by Vehicles (Podder, 2022), (2) Laptop Price Prediction using specifications (Chaki, 2023), (3) Used Car Price Prediction (Hinglaspure, 2024), and (4) Effects of Alcohol on Student Performance (Naude, 2024). Those datasets are selected based on three criteria: First, the dataset is publicly available dataset in CSV format that does not exceed 200KB in size and can be used for regression tasks. Second, the dataset contains issues that require cleaning. Third, the dataset has a usability rating of 10 on Kaggle, indicating its high value. Additional details regarding those datasets and corresponding cleaning challenges are provided in Appendix C.

To quantitatively measure agents' data cleaning ability and to guarantee evaluation fairness, we add constraints to the data cleaning task. The data cleaning task contains four subtasks designed based on the DC-RM procedure by Corrales et al. (2018) to evaluate agents' ability in data-driven programming, data analysis, reasoning, and instruction following. The subtasks are as follows:

- **Data Conversion:** The agent is tasked with converting all non-numerical columns to numerical form. Specifically, the agent must analyze the dataset and convert columns that contain numerical information but are stored as non-numerical data to numbers (e.g., "1.24 kg" to 1.24). In addition, Label (ordinal) encoding is used to convert all remaining categorical columns into numerical values.
- **Missing value imputation:** The agent is required to fill missing values (NaNs) using the Random Forest Imputation technique.
- **Outlier Detection and Removal:** The agent must identify and remove outliers using the Local Outlier Factor (LOF) method.
- **Duplicate Removal:** The agent must detect and remove duplicated rows in the dataset.

The task, along with detailed instructions for each subtask, is presented to agents either through a textual task description (for baseline agents) or via a SOP (for the SOP agents). For each method and dataset, we run the agent 10 times and report the average score. For the SOP-agent, we use an Act agent and the provided SOP can be found in Appendix H.

**Baselines** We use AutoGPT and MetaGPT's Data Interpreter (Hong et al., 2024) as baselines. AutoGPT is a general-purpose agent designed for a variety of tasks, including code generation. MetaGPT represents domain-specific agents and the state-of-the-art in solving data-driven problems.

**Evaluation Metrics** We evaluate the performance of the agent on the data-cleaning task by assessing the quality of the cleaned data using the following metrics:

- **remove_non_numeric_rate:** the percentage of cleaned data without non-numerical values.
- **remove_nan_rate:** the percentage of cleaned data with no missing values.
- **outlier_removal_rate:** the percentage of the cleaned data that contains fewer rows than the deduplicated original dataset.
- **remove_duplicate_rate:** the percentage of cleaned data without duplicated rows.
- **success_rate:** the percentage of cleaned data that contain neither non-numerical nor NaN values.

Among all metrics, the success_rate is particularly important because it indicates whether the data can be used for downstream tasks directly for most regression algorithms.

**Results and Observations** The results of the data cleaning task are visualized in Figure 4. The SOP agent achieves the best success rate of 100% which is significantly better than AutoGPT and competitive with Data Interpreter, a strong domain-specific agent in solving data-driven tasks.

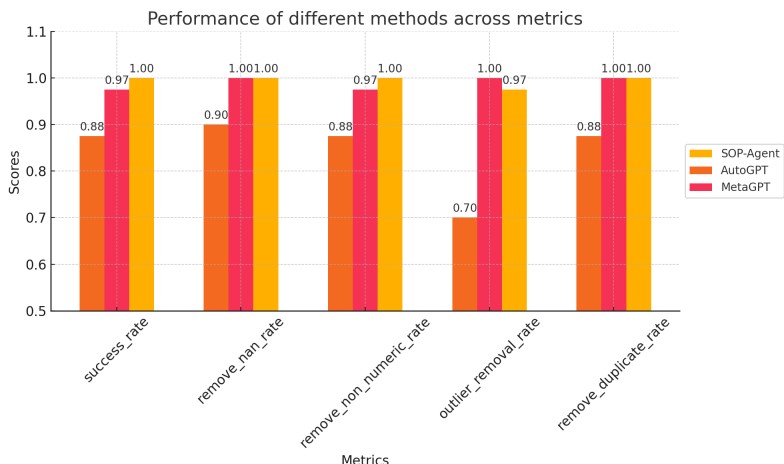

Figure 4: Results on the data-cleaning task

## 5  SOP Engineering

Techniques to improve prompting and tool calling performance have been widely discussed, including providing clear tool definitions and using few-shot examples. We further explore how to improve the stability of an SOP-based agent by engineering and rephrasing the SOP based on our empirical findings, refer to Appendix E for more details.

We find that with proper SOP setup, the SOP-agent can achieve extremely high performance in tasks that require intensive decision-making. We support our findings with the first SOP-grounded AI agent benchmark in custom service. This section will cover the benchmark data generation process, evaluation metrics, and final performance of SOP-agent.

### 5.1  Grounded Customer Service Benchmark

**The Task**   In industry, customer service providers need to provide assistance to customers according to a set of SOPs made by the company. They need to gather information from a variety of sources to assist decision making, such as querying an industrial database and asking the customers, or other related parties for clarification. They also need to perform actions, such as offering a refund, and escalate the issue to the corresponding team for addressal. Such tasks may not require high reasoning ability but demand high accuracy and robustness. Our benchmark is designed to simulate the customer service practice, where the agent plays the role of a customer service provider and acts based on SOPs in various use cases.

**Benchmark Data**   In the absence of an established benchmark dataset on evaluating AI agents' performance in grounded, decision-intensive tasks, we introduce the first customer service benchmark designed to assess the capability of AI agents in such settings. It covers customer service use cases across 5 different industries: online retail, food delivery services, ride-hailing services, telecommunications, and financial services (banking). For each industry, we write SOPs for 10 use cases of customer service practices. To simplify the benchmark, all function calls do not require arguments and there is no looping in the SOPs. Table 7 details on the statistics of the *Grounded Customer Service Benchmark*.

To automate the tests, we limit the output of function calls to three types: categorical (a string), boolean, and numerical. The benchmark provides a simulator, which simulates the observation for each function call by randomly selecting a value from a set of candidates specified in the testing data using rule-based algorithm (as detailed in Appendix D) to ensure extensiveness in the testing. For example, randomly generate a number from an auto-calculated range for numerical comparison.

For each industry, we ask GPT to generate 10 use cases where SOP may apply, such as "cancel a food order", "handle driver-customer conflict", and "make an appointment with a banker". Due

to the potential logical and coherence issues of LLM-generated content, such crude SOPs need to be manually refined to make sure that the SOP is logically intact and comprehensible to the LLM (GPT-4). We adopt the procedures as detailed in Algorithm 2 to manually refine the SOP. Refer to Appendix 2 for an example of the refined test case.

**Evaluation Metrics** We use two metrics to evaluate agents' performance on the proposed grounded customer service tasks: (1) path accuracy: a run is considered as successful if the called function calls match the ground truth one. (2) leaf accuracy: As a more lenient alternative to path accuracy, leaf accuracy focuses only on the outcomes of the function calls, only checking if all the leaf function calls (the last function call on that path) are called or not. Note that as not every function call provides essential information that may affect the decision-making process, some function calls are used to act without meaningful feedback, for example, "start refunding procedure". Missing such function calls won't affect the leaf node accuracy but will affect path accuracy greatly.

## 5.2 EXPERIMENTAL SETUP

For baselines, we use LangChain's (Chase & contributors, 2024) zero-shot ReAct agent. For both SOP-agent and the ReAct agent, we report the metrics based on 100 runs for each use case. To test the grounded task performance, we provide the SOP to the SOP-agent and a formatted textual SOP in bullet-point format to the baseline. All experiments use GPT-4 as the base model. As our dataset creation process inherently introduces biases if we attempt to use benchmarks on performance comparison, we include baselines in this experiment solely to identify any gaps that may need to be addressed to align existing agent systems with the complex, real-world challenges of grounding in customer service tasks.

## 5.3 RESULTS AND OBSERVATIONS

Table 5: Grounded Customer Service Benchmark Results

| Industry | ReAct (zero-shot) | | Ours | |
|---|---|---|---|---|
| | path_acc | leaf_acc | path_acc | leaf_acc |
| Online Retail | 77.10% | 82.50% | **100**% | **100**% |
| Food Delivery Services | 72.50% | 88.80% | **99.9**% | **99.9**% |
| Ride-Hailing Services | 75.90% | 84.07% | **99.8**% | **99.8**% |
| Telecommunications | 56.80% | 76.60% | **99.7**% | **99.7**% |
| Financial Services | 49.84% | 56.47% | **99.7**% | **99.7**% |
| **Average** | 67.43% | 77.68% | **99.8**% | **99.8**% |

As shown in Table 5, in our *Grounded Customer Service Benchmark*, the SOP-agent achieves extremely high scores in all categories, the overall accuracy is 99.8%. Meanwhile, the scores from the ReAct baseline suggest that the benchmark is still challenging for general-purpose AI agents.

## 6 CONCLUSION

In this work, we introduced SOP-agent, a novel autonomous agent system guided by pseudocode-style SOPs written in natural language to build task-specific agents. The SOP-agent addresses planning challenges in AI agents by guiding their behavior with predefined SOPs and dynamically adapting plans through selectively DFS traversal on a decision graph using function callings. We conducted extensive experiments across a variety of tasks, demonstrating the system's versatility. By incorporating human expertise, the SOP-agent offer better controllability and enable users without programming skills to define customized workflows through natural language SOPs. Experimental results show that the performance of the SOP-agent consistently outperforms general-purpose agent baselines and is comparable to strong domain-specific agents across multiple tasks, demonstrating both accuracy and robustness. However, limitations such as brittleness facing hallucination and the need for manually crafted SOPs remain. Despite these limitations, the SOP-agent give new inspiration for future research on autonomous AI agent systems in how to handle complex, real-world tasks.

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

## A ANALYSIS ON THEORETICAL LLM USAGE

There are three cases we need to consider, as shown in the figure below.

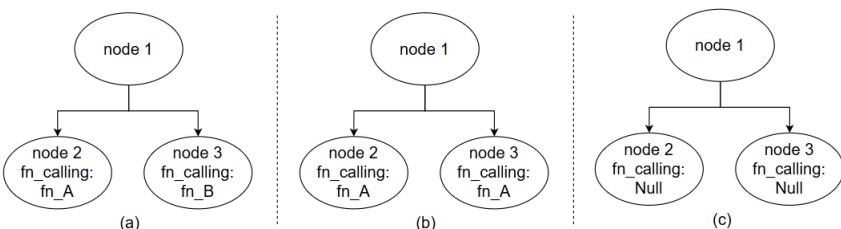

Figure 5: (a): **Subnodes have different function calls.** In this case, function calls made by the LLM can be used to determine the next branch to explore. For example, suppose we are currently at node 1; if 'fn_A' is called, the agent will explore node 2. (b): **Two or more function calls are the same.** In this case, we cannot tell which branch is chosen based on the function call made by the LLM, as both branches have the same function calls. (c): **At least two subnodes have no function calls.** This is the same as case (b). For both case (b) and case (c), we assign dummy function calls (e.g., "explore_subtree_A", "explore_subtree_B"...) to still be able to use function calling for branching, as in case (a). To generate the arguments for the function call, we use separate LLM calls for each chosen branch, which leads to $1 + |\text{branches\_applicable\_with\_function\_calling}|$ queries.

## B DETAILS ON USING DUMMY FUNCTIONS TO DO BRANCHING

For cases where the function calls cannot distinguish which branch to explore. We use dummy functions to do branching as described in the Algorithm 0.

---

**Algorithm 1** Branching with Dummy Function Callings

---

1: **Input:** Directed Graph $G$, Standard Operating Procedure (SOP), current node $N$
2: **Output:** The selected subnode of $N$, Execute the function call associated with the selected subnode, The observation if applicable.
3: **Initialize:** $S = \{s_1, s_2, \ldots, s_k \mid (N, s_k) \in G.edges\}$   $\triangleright s_k$ are those with edges from $N$ in $G$
4: **Initialize:** $D = \{\texttt{explore\_subtree\_A}, \texttt{explore\_subtree\_B}, \ldots\}$  $\triangleright$ k dummy function calls, one for each subnode
5: **if** $N$ has subnodes **then**
6:    Generate prompt $P$ by combining SOP and $D$     $\triangleright$ Generate a formatted prompt based on SOP and dummy functions
7:    $selected\_fn \leftarrow \text{LLM}(P)$        $\triangleright$ LLM selects most likely function based on the prompt
8:    $t \leftarrow \text{IndexOf}(selected\_fn, D)$
9:    $selected\_subnode \leftarrow S[t]$        $\triangleright$ Map the selected dummy function to the corresponding subnode
10:    $observation \leftarrow Null$
11:    **if** $selected\_subnode.fn\_call \neq$ Null **then**
12:       Generate arguments prompt $arg\_prompt$
13:       $arguments \leftarrow \text{LLM}(arg\_prompt)$   $\triangleright$ LLM generates arguments for the function call
14:       $observation \leftarrow calling\_with\_retry(selected\_subnode.fn\_call(arguments))$
15:    **end if**
16:    **return** $selected\_subnode, observation$
17: **else**
18:    **End** – No subnodes to explore.
19:    **return** $Null, Null$
20: **end if**

---

## C SUPPLEMENTARY DETAILS ABOUT THE DATA CLEANING TASK

**Data Cleaning Challenges in Datasets**   We select datasets that require different cleanups as listed in the table below.

| Dataset | Label Encoding | Remove Duplicated Rows | Regex-based Conversion | Remove NaN Values |
|---|---|---|---|---|
| CO2 Emission by Vehicles | ✓ | ✓ | ✗ | ✗ |
| Laptop Price Prediction using specifications | ✓ | ✗ | ✓ | ✓ |
| Used Car Price Prediction | ✓ | ✓ | ✗ | ✗ |
| Effects of Alcohol on Student Performance | ✓ | ✗ | ✓ | ✓ |

Table 6: Dataset Cleanup Requirements

## D SUPPLEMENTARY DETAILS ABOUT THE GROUNDED CUSTOMER SERVICE BENCHMARK

**Dataset Statistics**   The table above shows statistical metrics of the decision graph representation of SOPs in the grounded customer service benchmark. The average children per node metric calculates the average number of children for all non-leaf nodes.

**Random Test Case Generation**   Considering that most Standard Operating Procedures (SOPs) are unbalanced, merely exploring each sub-branch with equal probability can result in certain cases having a very slim chance of being explored. To address this, we have designed a balanced algorithm that randomly selects test cases to ensure fairness and extensiveness in our automated testing.

Table 7: Statistics of the Grounded Customer Service Dataset

| Statistic | Value |
|---|---|
| Average Maximum Depth | 4.52 |
| Number of Leaf Nodes | 6.18 |
| Number of Nodes | 11.82 |
| Number of Non-Leaf Nodes | 5.64 |
| Average Children per Node | 1.94 |
| Average Leaf Depth | 3.66 |
| Number of Unique APIs | 8.12 |

If the result of a function call A affects a set of precondition checks, we count the number of leaf nodes beneath the nodes of these precondition checks. We then randomly select a node to explore, with probabilities proportional to the number of leaf nodes in the sub-tree beneath that node. Finally, we generate random observations for the function call using a rule-based algorithm. For instance, if the function call returns a boolean value and two precondition checks use observations from the function call, and suppose the node corresponding to the first function has three leaf nodes in its sub-tree, while the other has one, our algorithm will generate "True" with a three-quarters chance and "False" with a one-quarter chance.

# E    ADDITIONAL RESULTS ON THE GROUNDED CUSTOMER SERVICE BENCHMARK

**Error Analysis**    We manually analyze the log of 9 failed cases from 5000 runs. Among failed cases, 3 runs failed because the LLM hallucinated a function-calling that doesn't exist. 6 of them are due to errors in reasoning, namely, the LLM chose a branch that should not be explored based on observation.

**SOP Refinement**    To ensure all the data samples in our constructed dataset is logically coherent and can be understood by GPT4, we adopt the following data refinement procedure (see Algorithm 2) to progressively fix the SOP through SOP engineering.

---
**Algorithm 2** Pseudocode for SOP refinement
---
1: **Input:** SOP
2: **while** True **do**
3:     need_manual_refine ← False
4:     **for** $i \in [0, 20]$ **do**
5:         set random seed to a random number
6:         reset test environment env
7:         trajectory ← sop_agent.run(sop)
8:         **if** trajectory == env.ground_truth_trajectory **then**
9:             **if** not success **then**
10:                 need_manual_refine ← True
11:             **end if**
12:         **end if**
13:     **end for**
14:     **if** need_manual_refine == True **then**
15:         manually refine the sop
16:     **else**
17:         **break**
18:     **end if**
19: **end while**
---

**Additional Study on SOP Engineering**    SOP engineering plays a crucial role in streamlining process optimization and enhancing workflow management efficiency. However, since those are out

of the scope of this paper, we will concentrate instead on how SOP engineering helps the SOP-agent system to improve its robustness. We found that carefully designed SOPs can help to improve the robustness of our proposed SOP-agent system by a large margin. The process involves checking the logical completeness of every logical chain in the SOP, using easy-to-understanding logic to avoid compound logic with "or" or "and", and matching function calling descriptions with action instructions in the SOP definition.

We demonstrate the process through a case study, in which we manually modify an SOP generated by the LLM to improve the SOP-agent's robustness. The crude SOP (see Listing 1), while used to guide the SOP-agent, achieves 84% in path accuracy based on 100 runs. We manually refine it to get the refined SOP (see Listing 2) and improve the path accuracy to 98%, which leads to a 16.7% improvement. The changed lines are presented in red in the refined SOP and the corresponding lines in the original crude SOP are in blue. The reason for making these modifications is to ensure the completeness of the logical chain. In the first modification, the precondition ("if the line is operational") cannot related to the previous function calling description ("Check the customer's, connection status") directly, which may introduce confusion and lead to sub-optimal performance. Similarly, in the second modification, the precondition (else if an interruption has been detected), although the previous function returns "'connection_status':'interruption_detected'", since the precondition didn't specify the scope of where it needs to find evidence regarding whether if an interruption has been detected, the LLM main attend to previous observation returned from the "check_area_outages" function call, which checks for any known outages in the customer's area and returns semantical similar responses ("'outage_status': 'outage reported'" and "'outage_status': 'outage none'").

Listing 1: Sample of Crude Example

```
- service_interruption_handling:
condition: "always"
API: {"name": "ServiceInterruptionHandle", "description": "
    ↪ Service Int. Handling SOP."}
Description: Customer reports service interruption
Instructions:
- authenticate customer's identity account details:
    condition: "always"
    API: {"name": "authenticate_customer", "description": "
        ↪ Confirm customer's identity and account details."}
    Instructions:
    - if account authentication fails, advise the customer to
        ↪ provide valid credentials or contact customer support
        ↪ for account recovery:
        condition: {"API": "authenticate_customer", "variable": "
            ↪ authentication_status", "condition_type": "is", "
            ↪ value": "failed"}
    - else if account authentication is successful, instantly
        ↪ verify the customer's account status.:
        condition: {"API": "authenticate_customer", "variable": "
            ↪ authentication_status", "condition_type": "is", "
            ↪ value": "success"}
        API: {"name": "verify_customer_account", "description": "
            ↪ Check the customer's account status."}
        Instructions:
        - if the account is inactive due to unpaid bills, advise
            ↪ the customer to make a payment and guide them
            ↪ through the payment process:
            condition: {"API": "verify_customer_account", "
                ↪ variable": "account_status", "condition_type": "
                ↪ is", "value": "inactive due to unpaid bill"}
        - else if the account is active, check for any known
            ↪ outages in the customer's area:
```

```
condition: {"API": "verify_customer_account", "
    ↪ variable": "account_status", "condition_type": "
    ↪ is", "value": "active"}
API: {"name": "check_area_outages", "description": "
    ↪ Check for any known outages in the customer's
    ↪ area."}
Instructions:
- if there is an outage, inform the customer of the
    ↪ outage and provide estimated time for resolution
    ↪ :
    condition: {"API": "check_area_outages", "variable
        ↪ ": "outage_status", "condition_type": "is", "
        ↪ value": "outage reported"}
    API: {"name": "check_outage_resolution_time", "
        ↪ description": "Provide an estimated time for
        ↪ when the service will be restored."}
    Instructions:
    - always apologize for the inconvenience and assure
        ↪  the customer that the company is working
        ↪ promptly to resolve the issue:
        condition: "always"
- else if there is no outages, proceed to
    ↪ troubleshooting and assess the customer's
    ↪ connection status:
    condition: {"API": "check_area_outages", "variable
        ↪ ": "outage_status", "condition_type": "is", "
        ↪ value": "none"}
    API: {"name": "assess_line_connection_status", "
        ↪ description": "Check the customer's
        ↪ connection status."}
    Instructions:
    - if the line is operational, guide the customer
        ↪ through a basic troubleshooting procedure
        ↪ based on interruption self-troubleshooting
        ↪ guide:
        condition: {"API": "
            ↪ assess_line_connection_status", "variable
            ↪ ": "connection_status", "condition_type":
            ↪ "is", "value": "operational"}
        API: {"name": "
            ↪ check_interruption_troubleshooting_guide",
            ↪  "description": "Check the interruption
            ↪ self-troubleshooting guide."}
        Instructions:
        - always ask the user if the problem is resolved
            ↪  or not:
            condition: "always"
            API: {"name": "
                ↪ query_problem_resolution_status", "
                ↪ description": "ask the customer if the
                ↪ problem is successfully resolved."}
            Instructions:
            - if problem is resolved, end the
                ↪ conversation politely:
                condition: {"API": "
                    ↪ query_problem_resolution_status", "
                    ↪ variable": "problem_status", "
                    ↪ condition_type": "is", "value": "
                    ↪ resolved"}
```

```
972                                  - else if the problem persists, escalate the
973                                  ↪ issue to technical support team:
974                                     condition: {"API": "
975                                        ↪ query_problem_resolution_status", "
976                                        ↪ variable": "problem_status", "
977                                        ↪ condition_type": "is", "value": "
978                                        ↪ persists"}
979                                  API: {"name": "
980                                        ↪ escalate_issue_to_technical_support
981                                        ↪ ", "description": "escalate the
982                                        ↪ issue to technical support team."}
983                          - else if an interruption has been detected,
984                             ↪ escalate the issue to the technical support
985                             ↪ team and open a service ticket:
986                              condition: {"API": "
987                                 ↪ assess_line_connection_status", "variable
988                                 ↪ ": "connection_status", "condition_type":
989                                 ↪ "is", "value": "interruption_detected"}
990                              API: {"name": "
991                                 ↪ escalate_issue_to_technical_support", "
992                                 ↪ description": "escalate the issue to
993                                 ↪ technical support team."}
```

Listing 2: Sample of Refined Example

```
- service_interruption_handling:
condition: "always"
API: {"name": "ServiceInterruptionHandle", "description": "
   ↪ Service Int. Handling SOP."}
Description: Customer reports service interruption
Instructions:
- authenticate customer's identity account details:
   condition: "always"
   API: {"name": "authenticate_customer", "description": "
      ↪ Confirm customer's identity and account details."}
   Instructions:
   - if account authentication fails, advise the customer to
      ↪ provide valid credentials or contact customer support
      ↪ for account recovery:
      condition: {"API": "authenticate_customer", "variable": "
         ↪ authentication_status", "condition_type": "is", "
         ↪ value": "failed"}
   - else if account authentication is successful, instantly
      ↪ verify the customer's account status.:
      condition: {"API": "authenticate_customer", "variable": "
         ↪ authentication_status", "condition_type": "is", "
         ↪ value": "success"}
      API: {"name": "verify_customer_account", "description": "
         ↪ Check the customer's account status."}
      Instructions:
      - if the account is inactive due to unpaid bills, advise
         ↪ the customer to make a payment and guide them
         ↪ through the payment process:
         condition: {"API": "verify_customer_account", "
            ↪ variable": "account_status", "condition_type": "
            ↪ is", "value": "inactive due to unpaid bill"}
      - else if the account is active, check for any known
         ↪ outages in the customer's area:
```

```
                    condition: {"API": "verify_customer_account", "
                       ↪ variable": "account_status", "condition_type": "
                       ↪ is", "value": "active"}
                 API: {"name": "check_area_outages", "description": "
                    ↪ Check for any known outages in the customer's
                    ↪ area."}
                 Instructions:
                 - if there is an outage, no troubleshooting is needed,
                    ↪  just inform the customer of the outage and
                    ↪ provide estimated time for resolution:
                    condition: {"API": "check_area_outages", "variable
                       ↪ ": "outage_status", "condition_type": "is", "
                       ↪ value": "outage reported"}
                    API: {"name": "check_outage_resolution_time", "
                       ↪ description": "Provide an estimated time for
                       ↪ when the service will be restored."}
                    Instructions:
                    - always apologize for the inconvenience and assure
                       ↪  the customer that the company is working
                       ↪ promptly to resolve the issue:
                       condition: "always"
                 - else if there is no outages, proceed to
                    ↪ troubleshooting and assess the customer's
                    ↪ connection status:
                    condition: {"API": "check_area_outages", "variable
                       ↪ ": "outage_status", "condition_type": "is", "
                       ↪ value": "none"}
                    API: {"name": "assess_line_connection_status", "
                       ↪ description": "Check the customer's
                       ↪ connection status."}
                    Instructions:
                    - if the connection status is 'operational', guide
                       ↪ the customer through a basic troubleshooting
                       ↪ procedure based on interruption self-
                       ↪ troubleshooting guide: %
                       condition: {"API": "
                          ↪ assess_line_connection_status", "variable
                          ↪ ": "connection_status", "condition_type":
                          ↪ "is", "value": "operational"}
                       API: {"name": "
                          ↪ check_interruption_troubleshooting_guide",
                          ↪  "description": "Check the interruption
                          ↪ self-troubleshooting guide."}
                       Instructions:
                       - always ask the user if the problem is resolved
                          ↪  or not:
                          condition: "always"
                          API: {"name": "
                             ↪ query_problem_resolution_status", "
                             ↪ description": "ask the customer if the
                             ↪ problem is successfully resolved."}
                          Instructions:
                          - if problem is resolved, end the
                             ↪ conversation politely:
                             condition: {"API": "
                                ↪ query_problem_resolution_status", "
                                ↪ variable": "problem_status", "
                                ↪ condition_type": "is", "value": "
                                ↪ resolved"}
```

```
1080                              - else if the problem persists, escalate the
1081                                ↪ issue to technical support team:
1082                                condition: {"API": "
1083                                    ↪ query_problem_resolution_status", "
1084                                    ↪ variable": "problem_status", "
1085                                    ↪ condition_type": "is", "value": "
1086                                    ↪ persists"}
1087                                API: {"name": "
1088                                    ↪ escalate_issue_to_technical_support
1089                                    ↪ ", "description": "escalate the
1090                                    ↪ issue to technical support team."}
1091                          - else if the connection status is '
1092                              ↪ interruption_detected', escalate the issue to
1093                              ↪  the technical support team and open a
1094                              ↪ service ticket: %
1095                            condition: {"API": "
1096                                ↪ assess_line_connection_status", "variable
1097                                ↪ ": "connection_status", "condition_type":
1098                                ↪ "is", "value": "interruption_detected"}
1099                          API: {"name": "
1100                              ↪ escalate_issue_to_technical_support", "
1101                              ↪ description": "escalate the issue to
                              ↪ technical support team."}
```

## F THE SOP USED IN THE ALFWORLD BENCHMARK

```
# zero-shot sops
- all in one:
condition_type: always
API: {"name": "AllInOne", "description": "Perform all tasks in
    ↪ the environment."}
Description: Perform all tasks in the environment.
Instructions:
- if the task is to put an object in/on somewhere, execute the
    ↪ plan 'pickup and place':
  API: pick_and_place
  condition_type: if
  Instructions:
  - list the places in obsearvation where the object to pickup
      ↪  can be located, order the list by possibility to find
      ↪  the object, start with the most likely place,
      ↪ checking all posible place one by one, start from the
      ↪ first place:
    API: go_to
    condition_type: always
    Instructions:
    - if the observation shows the place is an container and
        ↪ it is closed, open the container:
      API: open
      label: l03
      condition_type: if
      Instructions:
      - if object to pickup is in the container, take the
          ↪ object from the container:
        API: take
        condition_type: if
        goto: l02
```

```
          – else if object to pickup is not in the container, go
            ↪  to the next place to check for the object to
            ↪ pickup:
            API: go_to
            condition_type: if
            goto: l01, l03, l04
        – else if the object to pickup is in the location, take
          ↪ the object from the location:
          API: take
          label: l01
          condition_type: if
          Instructions:
          – if the observation shows the object to pickup has
            ↪ been taken, go to the place where you need to
            ↪ place the object:
          API: go_to
          label: l02
          condition_type: always
          Instructions:
          – if the observation shows the place is an
            ↪ container and it is closed, open the
            ↪ container:
          API: open
          condition_type: if
          Instructions:
          – if the observation shows the container is open
            ↪ , put the object in/on the place:
            API: put
            condition_type: if
        – if the observation shows a list of objects or
          ↪ nothing, put the object in/on the place:
          API: put
          condition_type: if
        – else if the object to pickup is not in the location or
          ↪ nothing happens, go to the next place to check for
          ↪ the object to pickup:
          API: go_to
          label: l04
          condition_type: if
          goto: l03, l01, l04
    – else if the task is to place a clean object it in/on
      ↪ somewhere, execute the plan 'pickup, clean, and place':
      API: pick_clean_and_place
      condition_type: if
      Instructions:
      – list the places in obsearvation where the object to clean
        ↪ can be located, order the list by possibility to find
        ↪ the object, start with the most likely place, checking
        ↪  all posible place one by one, start from the first
        ↪ place:
      API: go_to
      condition_type: always
      Instructions:
      – if the observation shows the place is an container and
        ↪ it is closed, open the container:
        API: open
        label: l13
        condition_type: if
        Instructions:
```

```
          - if exact object to clean is in the container, take
            ↪ the object from the container, you don't take an
            ↪  object if it is not the matching your target
            ↪ exactly:
            API: take
            condition_type: if
            goto: l12
          - else if object to clean is not in the container, go
            ↪ to the next place to check for the object to
            ↪ clean:
            API: go_to
            condition_type: if
            goto: l11, l13, l14
      - else if the exact object to clean is in the location,
        ↪ take the object from the location, you don't take
        ↪ an object if it is not the matching your target
        ↪ exactly:
        API: take
        label: l11
        condition_type: if
        Instructions:
        - always go to the sinkbasin to clean the object:
          API: go_to
          label: l12
          condition_type: always
          Instructions:
          - always clean the object:
            API: clean
            condition_type: always
            Instructions:
            - if the cleaning is successful, go to the place
              ↪  where you need to place the object:
              API: go_to
              label: l15
              condition_type: always
              Instructions:
              - if the observation shows the place is an
                ↪ container and it is closed, open the
                ↪ container:
                API: open
                condition_type: if
                Instructions:
                - if the observation shows the container
                  ↪ is open, put the object in/on the
                  ↪ place:
                  API: put
                  condition_type: if
                  Instructions:
                  - if the observation shows put is not
                    ↪ successful, make sure the action
                    ↪ is in correct format and try
                    ↪ again:
                    API: put
                    condition_type: if
              - if the observation shows a list of objects
                ↪ or nothing, put the object in/on the
                ↪ place:
                API: put
                condition_type: if
```

```
                                  Instructions:
                                  - if the observation shows put is not
                                      ↪ successful, make sure the action is
                                      ↪ in correct format and try again:
                                    API: put
                                    condition_type: if
                             - if the cleaning is not successful, make sure
                                 ↪ the action is in correct format and try
                                 ↪ again:
                                API: clean
                                label: l16
                                condition_type: if
                                goto: l15, l16
                  - else, go to the next place to check for the object to
                      ↪ clean:
                    API: go_to
                    label: l14
                    condition_type: if
                    goto: l13, l11, l14
        - else if the task is to place a hot object it in/on somewhere,
            ↪ execute the plan 'pickup, heat, and place':
          API: pick_heat_and_place
          condition_type: if
          Instructions:
          - list the places in obsearvation where the object to heat
              ↪ can be located, order the list by possibility to find
              ↪ the object, start with the most likely place, checking
              ↪  all posible place one by one, start from the first
              ↪ place:
            API: go_to
            condition_type: always
            Instructions:
            - if the observation shows the place is an container and
                ↪ it is closed, open the container:
              API: open
              label: l23
              condition_type: if
              Instructions:
              - if exact object to heat is in the container based on
                  ↪  observation, take the object from the container
                  ↪ :
                API: take
                condition_type: if
                goto: l22
              - else if object to heat is not in the container based
                  ↪  on observation, go to the next place to check
                  ↪ for the object to heat:
                API: go_to
                condition_type: if
                goto: l21, l23, l24
            - else if the object to heat is in the location, take the
                ↪  object from the location:
              API: take
              label: l21
              condition_type: if
              Instructions:
              - always go to microwave (as location) to heat the
                  ↪ object:
                API: go_to
```

```
label: l22
condition_type: always
Instructions:
- always heat the object, you can directly heat the
  ↪  object without any other action like open,
  ↪ put, close etc.:
  API: heat
  condition_type: always
  Instructions:
  - if the heating is successful, go to the place
    ↪ where you need to place the object:
    API: go_to
    label: l25
    condition_type: always
    Instructions:
    - if the observation shows the place is an
      ↪ container and it is closed, open the
      ↪ container:
      API: open
      condition_type: if
      Instructions:
      - if the observation shows the container
        ↪ is open, put the object in/on the
        ↪ place:
        API: put
        condition_type: if
        Instructions:
        - if the observation shows put is not
          ↪ successful, make sure the action
          ↪ is in correct format and try
          ↪ again:
          API: put
          condition_type: if
    - if the observation shows a list of objects
      ↪ or nothing, put the object in/on the
      ↪ place:
      API: put
      condition_type: if
      Instructions:
      - if the observation shows put is not
        ↪ successful, make sure the action is
        ↪ in correct format and try again:
        API: put
        condition_type: if
  - if the heating is not successful, make sure
    ↪ the action is in correct format and try
    ↪ again:
    API: heat
    label: l26
    condition_type: if
    goto: l25, l26
- else, go to the next place to check for the object to
  ↪ heat:
  API: go_to
  label: l24
  condition_type: if
  goto: l23, l21, l24
- else if the task is place a cool object in/on somewhere,
  ↪ execute the plan 'pickup, cool, and place':
```

```
API: pick_cool_and_place
condition_type: if
Instructions:
- list the places in obsearvation where the object to cool
    ↪ can be located, order the list by possibility to find
    ↪ the object, start with the most likely place, checking
    ↪  all posible place one by one, start from the first
    ↪ place:
  API: go_to
  condition_type: always
  Instructions:
  - if the observation shows the place is an container and
      ↪ it is closed, open the container:
    API: open
    label: l33
    condition_type: if
    Instructions:
    - if exact object to cool is in the container based on
        ↪  observation, take the object from the container
        ↪ :
      API: take
      condition_type: if
      goto: l32
    - else if object to cool is not in the container based
        ↪  on observation, go to the next place to check
        ↪ for the object to cool:
      API: go_to
      condition_type: if
      goto: l31, l33, l34
  - else if the exact object to cool is in the location
      ↪ based on observation, take the object from the
      ↪ location:
    API: take
    label: l31
    condition_type: if
    Instructions:
    - always go to the fridge (as location) to cool the
        ↪ object:
      API: go_to
      label: l32
      condition_type: always
      Instructions:
      - always cool the object, you can directly cool the
          ↪  object without any other action like open,
          ↪ put, close etc.:
        API: cool
        condition_type: always
        Instructions:
        - if the cooling is successful, go to the place
            ↪ where you need to place the object:
          API: go_to
          label: l35
          condition_type: always
          Instructions:
          - if the observation shows the place is an
              ↪ container and it is closed, open the
              ↪ container:
            API: open
            condition_type: if
```

```
                              Instructions:
                              - if the observation shows the container
                                ↪ is open, put the object in/on the
                                ↪ place:
                              API: put
                              condition_type: if
                              Instructions:
                              - if the observation shows put is not
                                  ↪ successful, make sure the action
                                  ↪ is in correct format and try
                                  ↪ again:
                              API: put
                              condition_type: if
                        - if the observation shows a list of objects
                            ↪ or nothing, put the object in/on the
                            ↪ place:
                          API: put
                          condition_type: if
                          Instructions:
                          - if the observation shows put is not
                              ↪ successful, make sure the action is
                              ↪ in correct format and try again:
                          API: put
                          condition_type: if
                    - if the cooling is not successful, make sure
                        ↪ the action is in correct format and try
                        ↪ again:
                      API: heat
                      label: l36
                      condition_type: if
                      goto: l35, l36
        - else, go to the next place to check for the object to
            ↪ cool:
          API: go_to
          label: l34
          condition_type: if
          goto: l33, l31, l34
  - else if the task is to look at some object under a desklamp,
      ↪ execute the plan 'look at':
    API: pick_and_look
    condition_type: if
    Instructions:
    - list the places in obsearvation where the object to look
        ↪ at (other than the desklamp) can be located, order the
        ↪  list by possibility to find the object, start with
        ↪ the most likely place, checking all posible place one
        ↪ by one, start from the first place:
      API: go_to
      condition_type: always
      Instructions:
      - if the observation shows the place is an container and
          ↪ it is closed, open the container:
        API: open
        label: l43
        condition_type: if
        Instructions:
        - if exact object to look at (other than the desklamp)
            ↪  is in the container based on observation, take
            ↪ the object from the container:
```

```
                        API: take
                        condition_type: if
                        goto: l42, l48
                  - else if object to look at (other than the desklamp)
                    ↪ is not in the container based on observation, go
                    ↪  to the next place to check for the object to
                    ↪ look at:
                        API: go_to
                        condition_type: if
                        goto: l43, l41, l44, l49
              - else if the exact object to look at (other than the
                ↪ desklamp) is in the location based on observation,
                ↪ take the object from the location:
                API: take
                label: l41
                condition_type: if
                Instructions:
                - if you already saw the desklamp somewhere, go to the
                  ↪  place where you saw the desklamp:
                    API: go_to
                    label: l42
                    condition_type: if
                    goto: l45, l46, l47
                - else if the desklamp is not found yet. List the
                  ↪ places in environment where a desklamp can be
                  ↪ located, order the list by possibility to find
                  ↪ the desklamp, go to the most likely place,
                  ↪ checking all posible place one by one:
                    API: go_to
                    label: l48
                    condition_type: if
                    Instructions:
                    - if the observation shows the place is an
                      ↪ container and it is closed, open the
                      ↪ container:
                        API: open
                        label: l45
                        condition_type: if
                        Instructions:
                        - if desklamp is in the container, use the
                          ↪ desklamp:
                            API: use
                            condition_type: if
                        - else if desklamp is not in the container, go
                          ↪ to the next place to check for the object
                          ↪ to look at:
                            API: go_to
                            condition_type: if
                            goto: l45, l46, l47
                    - else if the desklamp is in the location, use the
                      ↪ desklamp:
                        API: use
                        label: l46
                        condition_type: if
                    - else if the observation shows the desklamp is not
                      ↪  in the location, go to the next place to
                      ↪ check for the desklamp:
                        API: go_to
                        label: l47
```

```
                        condition_type: if
                        goto: l45, l46, l47
            - else if the desklamp is in the location based on the
              ↪ observation but the object to look at is not found,
              ↪  go to the next place to check for the object to
              ↪ look at:
              API: go_to
              label: l44
              condition_type: if
              goto: l43, l41, l44, l49
            - else if the object to look at is not in the location or
              ↪  nothing happens, go to the next place to check for
              ↪  the object to look at:
              API: go_to
              label: l49
              condition_type: if
              goto: l43, l41, l44, l49
    - else if the task is to place two objects in/on somewhere,
      ↪ execute the plan 'pickup and place twice':
      API: pick_and_place_two
      condition_type: if
      Instructions:
      - list the places in obsearvation where the object to pickup
        ↪  can be located, order the list by possibility to find
        ↪  the object, start with the most likely place,
        ↪ checking all posible place one by one, start from the
        ↪ first place:
        API: go_to
        condition_type: always
        Instructions:
        - if the observation shows the place is an container and
          ↪ it is closed, open the container:
          API: open
          label: l53
          condition_type: if
          Instructions:
          - if exact object to pickup is in the container based
            ↪ on the observation, take the object from the
            ↪ container:
            API: take
            condition_type: if
            goto: l52
          - else if object to pickup is not in the container
            ↪ based on the observation, go to the next place
            ↪ to check for the object to pickup:
            API: go_to
            condition_type: if
            goto: l53, l51, l54
        - else if the exact object to pickup is in the location
          ↪ based on the observation, take the object from the
          ↪ location:
          API: take
          label: l51
          condition_type: if
          Instructions:
          - go to the place or object (as location) where you
            ↪ need to place the object:
            API: go_to
            label: l52
```

```
                    condition_type: always
                    Instructions:
                    - if the observation shows the place is an
                      ↪ container and it is closed, open the
                      ↪ container:
                      API: open
                      condition_type: if
                      Instructions:
                      - if the observation shows the container is open
                        ↪ , put the object in/on the place:
                        API: put
                        condition_type: if
                        Instructions:
                        - if you already saw the second object
                          ↪ somewhere, go to the place where you
                          ↪ saw the second object:
                          API: go_to
                          condition_type: if
                          goto: l53, l51, l54
                        - else, list the rest places in environment
                          ↪ where you can find the second object
                          ↪ and have not visited, start with the
                          ↪ most likely place, checking all posible
                          ↪  place one by one, start from the first
                          ↪  place:
                          API: go_to
                          condition_type: if
                          goto: l53, l51, l54
                  - else if the observation shows a list of objects
                      ↪ or nothing, put the object in/on the place:
                      API: put
                      condition_type: if
                      Instructions:
                      - if you already saw the second object somewhere
                        ↪ , go to the place where you saw the second
                        ↪  object:
                        API: go_to
                        condition_type: if
                        goto: l53, l51, l54
                      - else, list the rest places in environment
                        ↪ where you can find the second object and
                        ↪ have not visited, start with the most
                        ↪ likely place, checking all posible place
                        ↪ one by one, start from the first place:
                        API: go_to
                        condition_type: if
                        goto: l53, l51, l54
            - else, go to the next place to check for the object to
                ↪ pickup:
              API: go_to
              label: l54
              condition_type: if
              goto: l53, l51, l54
```

## F.1 APPENDIX B: THE SOP USED IN THE HOTPOTQA BENCHMARK

```
- multihop-question-answering-react:
    condition_type: always
```

```
API: {"name": "MultiHopQA", "description": "Generate code given
    ↪ the description."}
Description: Multi-hop QA SOP
Instructions:
- think about what to do next based on the provided question
    ↪ and answer and obtained information. log your thought to
    ↪ memory with key 'thought':
  API: log_thought
  label: think
  condition_type: always
  Instructions:
  - Evaluate the change for the key information to appear in
      ↪ the article whose first paragraph is the last
      ↪ observation, if the change is high, lookup for
      ↪ keywords in the article with the lookup tool,
      ↪ otherwise search for a different entity with the
      ↪ search tool:
    API: action_selection
    label: action_selection
    condition_type: always
    Instructions:
    - if search is the next action to perform, search the
        ↪ Wikipedia for an entity (name of person/object) to
        ↪ obtain a new article related to the entity, you
        ↪ should avoid searching for the same entity multiple
        ↪ times:
      API: search_new_article
      label: search
      condition_type: if
      Instructions:
      - always, log the key information in the result, if
          ↪ the search cannot find the entity, log the
          ↪ similar entities:
        API: log_result
        condition_type: always
        Instructions:
        - always, think about what action to take next. log
            ↪ your thought.:
          API: log_thought
          condition_type: always
          Instructions:
          - if the question is answerable, answer the
              ↪ question with very short response (either
              ↪ yes or no or a the name of the entity, a
              ↪ number, etc.), note that every question is
              ↪ guaranteed to have a valid answer:
            API: answer
            condition_type: if
          - else, search for more information:
            API: search_more_information
            condition_type: if
            goto: action_selection
    - if lookup is the next action to perform, lookup for
        ↪ certain keywords in the current file to obtain more
        ↪ information that help to answer the question:
      API: lookup_keyword_in_current_article
      label: lookup
      condition_type: if
      Instructions:
```

```
                    - always, log the key information in the result:
                      API: log_result
                      condition_type: always
                      Instructions:
                      - always, think about what action to take next. log
                        ↪  your thought.:
                      API: log_thought
                      condition_type: always
                      Instructions:
                      - if the question is answerable, answer the
                        ↪ question with very short response (either
                        ↪ yes or no or a the name of the entity, a
                        ↪ number, etc.), note that every question is
                        ↪  guaranteed to have a valid answer:
                      API: answer
                      condition_type: if
                    - else, search for more information:
                      API: search_more_information
                      condition_type: if
                      goto: action_selection
```

## G  SOP USED IN CODE GENERATION

```
- simple_code_generation:
condition_type: always
API: {"name": "CodeGen", "description": "Generate code given
    ↪ the description."}
Description: Code generation SOP
Instructions:
- Think about the problem and try to understand the
    ↪ requirements. Generate a plan to solve the problem. Also,
    ↪  explain at least one test cases step by step. add an
    ↪ entry to the memory with key 'thought' to log your
    ↪ thought with key.:
  API: log_to_memory
  condition_type: always
- Initialize a retry_counter with value 0, add an entry to the
    ↪ memory with key 'retry_counter', use `retry_counter = XX
    ↪ `:
  API: log_to_memory
  condition_type: always
- Generate a python function along with a unit test that
    ↪ contains test cases in a single file, add an entry to the
    ↪  memory with key 'code' to record the program and the
    ↪ unit tests in plain text:
  API: log_to_memory
  condition_type: always
- Execute the generated program stored in memory with the key '
    ↪ code' using python.:
  API: python
  condition_type: always
  Instructions:
  - If retry_counter<4 and there is any error message appears
      ↪ in the python code execution results, explain the
      ↪ error and provide suggestions on how to revise the
      ↪ code, update the 'thought' entry of the memory with
      ↪ your thought:
    API: log_to_memory
```

```
1728            condition_type: if
1729            label: retry_loop_start
1730            Instructions:
1731            - Increate the retry_counter by 1, update the '
1732              ↪ retry_counter' entry in memory:
1733             API: log_to_memory
1734             condition_type: always
1735            - Fix or rewrite the previous generated code and unit
1736              ↪ tests in the memory based on the thought and the
1737              ↪ error message, update the 'code' entry in memory
1738              ↪ with the new code:
1739             API: log_to_memory
                 condition_type: always
1740            - Execute the generated program stored in memory with the
1741              ↪  key 'code' using python:
1742             API: python
1743             condition_type: always
1744             goto: retry_loop_start, retry_loop_end
1745          - If the retry_counter>=4 or the code passed all unit tests,
1746            ↪  save your code:
1747           API: save_code
1748           condition_type: if
1749           label: retry_loop_end
1750
1751
```

## H  SOP USED IN DATA CLEANING

```
1753    - regression_data_cleaning:
1754    condition_type: always
1755    API: {"name": "DataCleaning", "description": "Data cleaning SOP
1756        ↪ ."}
1757    Description: Data cleaning SOP
1758    Instructions:
1759    - write code to 1. read data from data.csv, 2. check the data
1760        ↪ types of all columns, print the result:
1761      API: python
1762      condition_type: always
1763      Instructions:
1764      - log the data types of all columns to memory with the key "
1765          ↪ data_types":
1766       API: log_to_memory
1767       condition_type: always
1768       Instructions:
        - write code or fix code to 1. read data from data.csv,
1769          ↪ 2. convert all non-numerical columns to numerical
1770          ↪ columns with ordinal (label) encoding, 3. save the
1771          ↪ processed data to data_numerical.csv:
1772         API: python
1773         condition_type: always
1774         label: convert_categorical_to_numerical
1775         Instructions:
         - if the previous step failed, retry previous step:
1776           condition_type: if
1777           goto: convert_categorical_to_numerical
1778         - else, write code or fix code to 1. read data from
1779           ↪ data_numerical.csv, 2. check if all columns are
1780           ↪ numerical, print the result:
1781           API: python
             label: check_numerical_columns
```

```
condition_type: if
Instructions:
- if previous step failed, retry previous step:
    condition_type: if
    goto: convert_categorical_to_numerical
- else if not all columns are numerical, retry
    ↪ converting non-numerical columns to numerical
    ↪  columns:
    condition_type: if
    goto: convert_categorical_to_numerical
- else, write code or fix code to 1. read data from
    ↪  data_numerical.csv, 2. fill NaN values with
    ↪ random forest imputation, 3. save the
    ↪ processed data back to data_impute.csv:
    API: python
    label: fill_nan
    condition_type: if
    Instructions:
    - if previous step failed, retry previous step:
        condition_type: if
        goto: fill_nan
    - else, write code or fix code to 1. read data
        ↪ from data_impute.csv, 2. check if there is
        ↪  NaN values in the data, print the result:
        API: python
        label: check_nan_values
        condition_type: if
        Instructions:
        - if previous step failed, retry previous
            ↪ step:
            condition_type: if
            goto: fill_nan
        - else if there is still a NaN value in the
            ↪ data, retry filling NaN values with
            ↪ random forest imputation:
            condition_type: if
            goto: fill_nan
        - else, write code or fix code to 1. read
            ↪ data from data_impute.csv, 2. detect
            ↪ and remove outliers with local outlier
            ↪ factor method, 3. save the processed
            ↪ data back to data_remove_outlier.csv:
            API: python
            condition_type: always
            label: remove_outliers
            Instructions:
            - if previous step failed, retry previous
                ↪ step:
                condition_type: if
                goto: remove_outliers
            - else, write code or fix code to 1. read
                ↪ data from data_remove_outlier.csv,
                ↪ 2. remove duplicated rows, 3. save
                ↪ the processed data back to
                ↪ data_deduplicated.csv:
                API: python
                condition_type: always
```

