# OpenReview forum: "SOP-Agent: Empower General Purpose AI Agent with Domain-Specific SOPs"
_ICLR.cc/2025/Conference — ICLR 2025 Conference Withdrawn Submission_

### Official Review · Reviewer_Qsc7 · 2024-11-01

**Soundness:** 3
**Presentation:** 2
**Contribution:** 2
**Rating:** 3
**Confidence:** 4

**Summary:**

This paper proposes SOP-Agent, a framework that enables an LLM-based agent to leverage domain-specific knowledge by representing SOPs as decision graphs and conveying them to the LLM in a pseudo-code text format. The authors validate SOP-Agent across various benchmarks and introduce a new benchmark specifically designed for customer service scenarios.

**Strengths:**

- The concept of using SOPs to enable an LLM-based agent to effectively utilize domain-specific knowledge is well-founded.
- The authors validate the SOP-Agent across diverse benchmarks.
- The introduction of a Grounded Customer Service Benchmark targets a practical application area and could be valuable for further research into LLM-based customer service solutions.

**Weaknesses:**

- The primary limitation of SOP-Agent is that, while the paper critiques existing domain-specific agents for relying on hardcoded, human-designed workflows, SOP-Agent itself is subject to the same limitation. SOP-Agent still requires human expertise to design the decision graph, which can become large and challenging in complex domains.
- The explanation of SOP-Agent is insufficient. For example, the interaction between the decision graph and the LLM agent lacks clarity, including how SOP-Agent determines which node to execute, how it evaluates condition satisfaction after each action, what input is provided to the LLM agent at each step, and how the model handles the generation of actions not included in the decision graph.
- The fixed decision graph approach appears similar to behavior tree methods, such as MOSAIC [1]. To highlight the novelty of SOP-Agent, it would be helpful to explain how it differs from these existing methods.
- The use of a predefined decision graph limits adaptability, making it difficult for SOP-Agent to handle unexpected situations.
- There is a concern that the Data Cleaning experiment could not be appropriate for evaluating LLM agent performance.
- The performance gap between SOP-Agent and ReAct is minimal in the ALFWorld and HotpotQA experiments, and in the code generation task, SOP-Agent performs worse than some other algorithms. Justification for SOP-Agent’s advantages over these methods is needed.
- Although the motivation behind the proposed Grounded Customer Service Benchmark is sound, the benchmark’s difficulty appears relatively low. With a maximum depth of 4.52 and an average of 1.94 child nodes per node, as evidenced by SOP-Agent’s high performance (99.8%).
- *(Minor)* The citation format should be improved.
- *(Minor)* Using mathematical notation for performance metrics would enhance readability.

[1] Wang, Huaxiaoyue, et al. "MOSAIC: A Modular System for Assistive and Interactive Cooking." *arXiv preprint arXiv:2402.18796* (2024).

**Questions:**

- What are the advantages of SOP-Agent’s decision graph design compared to MetaGPT’s multi-agent SOP setup? (In other words, what benefits does designing SOPs as a decision graph offer over the multi-agent approach used by MetaGPT?)
- How does the decision graph approach differ from behavior trees, such as those used in MOSAIC [1]?
- Can the input token counts for SOP-Agent, ReAct, Auto-GPT, and MetaGPT be compared to assess efficiency?
- In the code generation experiments, could more detail be provided about the use of long-term memory? Additionally, since it is unclear whether the observed performance is due to memory or SOP-Agent itself, could ablation study results be shared?
- Do the authors believe the Grounded Customer Service Benchmark is sufficiently challenging? Additionally, how would ReAct perform in a few-shot setting, and are SOP-Agent’s results based on a zero-shot setup?

---

### Official Review · Reviewer_gLvk · 2024-11-02

**Soundness:** 3
**Presentation:** 2
**Contribution:** 2
**Rating:** 3
**Confidence:** 3

**Summary:**

This paper introduces the SOP-agent framework, an innovative artificial intelligence agent system that integrates SOPs into natural language processing to guide agents in executing complex tasks. The SOP-agent employs decision graphs to represent SOPs, enabling agents to dynamically adapt to their environments and make informed decisions through a DFS strategy. Extensive experimental validation across various domains, including decision-making, interactive search, code generation, and data cleaning, demonstrates the framework's superior performance and versatility.

**Strengths:**

1. The SOP-agent framework introduces an innovative approach that integrates SOPs with LLMs to enhance the task execution capabilities of AI agents in specialized domains.
2. This paper demonstrates the application of SOP-agent across various fields, including decision-making, search reasoning, code generation, and data cleansing, thereby establishing its extensive applicability and multifunctionality.

**Weaknesses:**

1. The SOP-agent employs decision graphs to represent Standard Operating Procedures (SOPs). However, it appears that the creation of these decision graphs necessitates substantial input from domain experts. Furthermore, it may be more appropriate to refer to these as decision trees rather than decision graphs.
2. The primary contribution of this paper lies in the integration of SOPs into agent systems through the use of decision graphs. However, as noted in related works, agent methodologies based on SOPs have already been explored.
3. The methodology section of the paper is overly simplistic. It is essential to provide more detailed information regarding the SOP-agent framework. For instance, in the specified decision graph, is the reasoning of the LLM constrained in any way? Can the approach be guaranteed to maintain adaptability to varying environments, as well as scalability and generalization capabilities?
4. Mirror: the formatting of tables should be standardized throughout the document.

**Questions:**

Please refer to the section of Weaknesses.

---

### Official Review · Reviewer_pw26 · 2024-11-05

**Soundness:** 2
**Presentation:** 2
**Contribution:** 2
**Rating:** 3
**Confidence:** 4

**Summary:**

Current general-purpose AI agents face challenges with long-horizon planning and effectively utilizing domain-specific knowledge, limiting their real-world applications. To address this issue, this paper proposes a SOP-agent framework, which addresses these limitations by using pseudocode-style Standard Operational Procedures written in natural language, represented as decision graphs that guide task completion. Empirical evaluation across multiple domains show that SOP-agent achieves better performance than general-purpose agents while matching domain-specific systems. The work also introduces the Grounded Customer Service Benchmark to evaluate AI agents' decision-making capabilities in SOP-based customer service scenarios.

**Strengths:**

1. The work is addressing a relevant issue, so problem itself is unique.
2. The evaluations is done on a lot of diverse domains, so shows that with proper tweakingm the method can be widely used.

**Weaknesses:**

Lack of solution novelty. The authors explain the method in very short with many missing details (see questions), and then go on and on about the benchmarks and baselines. Probably this work was suitable for the Dataset and Benchmark track.

**Questions:**

1. Who provides the SOP Navigator in the form of Decision Tree?

2. Line 173: we provide the agent with a filtered set of valid function callings (see Figure 1). What is the “filtered set of valid function callings” referred to here? And how is it provided?

3. Lines 180-181: “the corresponding function (if any) will be executed with parameters generated by separate LLM calls”. How do you verify the accuracy of the lLM generated parameters? If no, how do you handle cases if the parameters are inaccurate?

4. Line 182: What is the variable $branches$\_$with$\_$function$_$calls$? This is the first instance of its uage, and its meaning is assumed to be trivial for readers to understand.

5. Lines 183-185: “In this case, the function callings that the agent made help determine which conditions are met, allowing for more efficient branching”. Can you please provide an example of this? This sentence is not clear at all for anyone not familiar with the work.

6. Lines 261-264: Thanks authors for investigating the issue. I still feel that this is a major shortcoming of the approach. SOPs are supposed to be followed and the whole paper was building on that. Can an LLM ignore the SOP? If yes, how often does the LLM ignore the SOP, and what measures could be implemented to ensure better adherence to the SOP?

---

### Official Review · Reviewer_34i9 · 2024-11-07

**Soundness:** 2
**Presentation:** 2
**Contribution:** 2
**Rating:** 3
**Confidence:** 4

**Summary:**

The paper introduces a novel approach for enhancing general-purpose AI
agents through Standard Operational Procedures (SOPs) formatted as decision
graphs. By utilizing pseudocode-style SOPs, the authors aim to improve the
agent’s ability to perform domain-specific tasks, such as customer service,
data cleaning, and decision-making, which often require robust task-specific
workflows. The paper presents a new "Grounded Customer Service
Benchmark" for evaluating AI agents in decision-intensive settings and
demonstrates the versatility and effectiveness of SOP-based guidance
compared to general-purpose and specialized agents.

**Strengths:**

1. Originality: The SOP-agent framework is innovative, as it leverages structured
SOPs to guide AI agents, enabling better task-specific performance while
maintaining the adaptability of general-purpose agents. This approach
addresses a gap in developing flexible but specialized agents. The experiments
are extensive, covering multiple domains, including decision-making, multi-hop
question answering, and data cleaning. The use of SOPs appears to reduce
error rates and improve task accuracy across diverse settings.

2. Clarity: The methodology and implementation of SOP guidance using decision
graphs are clearly described, supported by both visual and quantitative results.
Significance: The Grounded Customer Service Benchmark provides a valuable
dataset for assessing the effectiveness of AI agents in realistic scenarios,
potentially contributing significantly to future research in AI-based customer
service and other real-world applications.

**Weaknesses:**

1. Novelty: While SOP guidance for AI agents is an innovative approach, the
contribution could be seen as incremental, primarily advancing on the design of
domain-specific SOPs rather than fundamentally new techniques.
Experimental Limitations: The paper could benefit from broader comparisons
with additional, more varied baseline models. Additionally, it lacks a detailed
analysis of failure cases in the Grounded Customer Service Benchmark, which
might provide insights into areas needing further improvement.

2. Robustness: The framework’s reliance on manually crafted SOPs may reduce
scalability, as these SOPs might need frequent updates for new domains or
changing environments, impacting the agent’s adaptability.

**Questions:**

1. Could the authors clarify how well the SOP-agent generalizes when minor
modifications are made to SOPs? This information would be valuable in
understanding the framework’s adaptability.

2. Could the authors provide a more detailed error analysis for the tasks where
SOP-agent underperformed compared to baselines, particularly in customer
service scenarios?

3. How does the SOP-agent handle unexpected task flows or out-of-scope
inputs that are not predefined in the SOP?

---

### Author Response · Authors · 2024-11-15
**Withdraw Comment**

We thank the reviewers for reviewing our paper. After careful consideration, we think our paper is inappropriate for ICLR, and we decided to withdraw our paper.

---

### Note · Authors · 2024-11-15

I have read and agree with the venue's withdrawal policy on behalf of myself and my co-authors.